# Scaling Cross-Embodied Learning: One Policy for Manipulation, Navigation, Locomotion and Aviation

**Ria Doshi**[*1]    **Homer Walke**[*1]    **Oier Mees**[1]    **Sudeep Dasari**[2]    **Sergey Levine**[1]

[1]UC Berkeley    [2]Carnegie Mellon University

https://crossformer-model.github.io

**Abstract:** Modern machine learning systems rely on large datasets to attain broad generalization, and this often poses a challenge in robot learning, where each robotic platform and task might have only a small dataset. By training a single policy across many different kinds of robots, a robot learning method can leverage much broader and more diverse datasets, which in turn can lead to better generalization and robustness. However, training a single policy on multi-robot data is challenging because robots can have widely varying sensors, actuators, and control frequencies. We propose CrossFormer, a scalable and flexible transformer-based policy that can consume data from any embodiment. We train CrossFormer on the largest and most diverse dataset to date, 900K trajectories across 20 different robot embodiments. We demonstrate that the same network weights can control vastly different robots, including single and dual arm manipulation systems, wheeled robots, quadcopters, and quadrupeds. Unlike prior work, our model does not require manual alignment of the observation or action spaces. Extensive experiments in the real world show that our method matches the performance of specialist policies tailored for each embodiment, while also significantly outperforming the prior state of the art in cross-embodiment learning.

**Keywords:** Imitation Learning, Cross-Embodiment

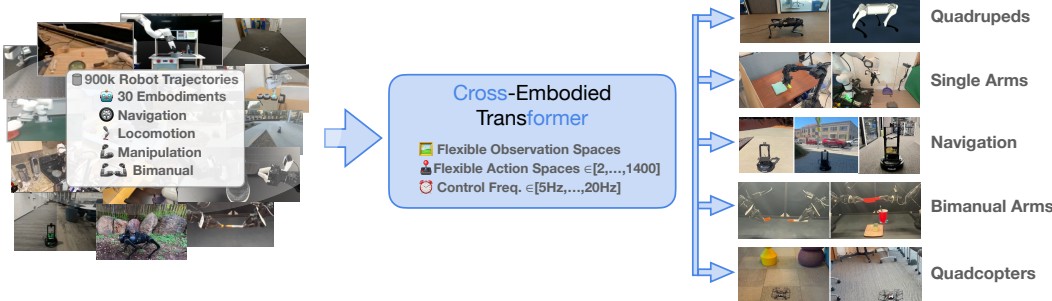

Figure 1: We introduce CrossFormer, a transformer-based policy trained on 900K trajectories of diverse, multi-embodied robot data that can control vastly different robots including single and dual arm manipulation systems, wheeled robots, quadcopters, and quadrupeds, while matching the performance of specialist policies targeted to each embodiment and outperforming prior work in cross-embodiment learning.

## 1 Introduction

Much of the recent success in machine learning has been driven by training general-purpose models on increasingly diverse and multi-task data. For example, visual and language tasks, once handled by task-specific methods, are now performed more effectively by general vision-language models that can transfer knowledge across tasks [1, 2, 3, 4]. Similarly, in robotics, recent data aggregation efforts [5] have made it possible to train general-purpose policies on robot data collected across

---

[*]Equal contribution. Corresponding email: {`riadoshi, homer_walke`}@berkeley.edu

8th Conference on Robot Learning (CoRL 2024), Munich, Germany.

multiple embodiments, tasks, and environments. These generalist policies can out-perform narrow policies trained with data from only the target robot and task by transferring visual representations and skills [6, 5]. In addition to the benefits of positive transfer, training generalist cross-embodied policies minimizes the amount of engineering that must be done to design and tune policy architectures for each robot.

However, training a general-purpose robot policy is uniquely challenging, since robot systems can vary widely in their camera views, proprioceptive inputs, joint configurations, action outputs, and control frequencies. Initial efforts at training large-scale cross-embodied policies have often been limited to single robot arms or ground navigation robots which can be controlled with a single camera view and relative waypoint action for the base or end-effector [5, 6, 7, 8]. Further increasing the diversity of embodiments these policies can control requires a model architecture that supports conditioning on any number of camera views or proprioceptive observations, as well as predicting actions of any dimension. Following prior work, we take a sequential modeling approach to cross-embodied imitation learning [9, 10]. We propose a transformer-based policy that supports variable observations and actions by casting inputs and outputs to a sequence. We scale this approach to control the most diverse set of embodiments with a single policy to date, including single and bimanual robot arms, ground navigation robots, quadcopters and quadrupeds.

With our transformer policy, we can train on robot data with any number of camera views or proprioceptive sensors by simply tokenizing and arranging the observations into a sequence. At the same time, we can predict actions of any dimension, crucially without the need to manually align the action spaces of different embodiments [8]. For each action type, we insert a set of *action readout tokens* into the input token sequence. Then, we pass the corresponding output embeddings into action-space specific heads to produce a vector of the correct dimensions. Our policy can accept tasks in the form of language instructions or goal images, allowing a user to choose the task modality that is most natural for a given embodiment.

Our primary contribution is a cross-embodied robot policy trained on the largest and most diverse robot dataset to date with 900K trajectories and 20 distinct embodiments. Our policy can control robots with varying observation and action types, from a quadruped with proprioceptive sensors and 12 joints to a bimanual robot with 3 cameras and 14 joints. We find in extensive real world experiments that our policy matches the performance of the same architecture trained on just the target robot data, as well as the best prior method in each setting, demonstrating that our architecture can absorb heterogeneous robot data without negative transfer, while performing comparably to state-of-the-art specialist methods tailored for each robot. Additionally, we find that our method outperforms the state of the art in cross-embodiment learning while alleviating the need for manually aligning observation and action spaces.

## 2 Related Work

Early work on cross-embodied robot policy learning has explored a number of techniques, including conditioning on explicit or learned representations of the embodiment, [11, 12], domain randomization and adaptation [13, 14, 15, 16, 17, 18], modular policies [19, 20, 21, 22], or model-based RL [23, 24, 25]. Generally, these prior projects have operated on a smaller scale, only evaluating in simulation or training policies on small amounts of robot data to control a small number of robots for a few tasks.

A number of prior works have sought to scale up robot learning by using large amounts of data from a single robot embodiment, collected either autonomously [26, 27, 28, 29, 30, 31] or with human teleoperation [32, 33, 34, 35, 36, 37, 38, 39]. Other prior works train on data from multiple robots, but require each robot to have the same observation and action space [7, 40, 5, 41, 42, 43, 44]. For example, Shah et al. [42] train one policy across many navigation robots using an egocentric camera view and 2D waypoint actions, and the RT-X models [5] are trained across single robotic arms using a 3rd person camera view and 7-DoF end-effector position actions. Our method does not require the data to have a common observation and action space, enabling simultaneous control of robots with disjoint sets of sensors and actuators, e.g., robot arms and quadrupeds.

There have been a few large-scale efforts to train a single policy on robot data with varying observations and action spaces [10, 9, 6, 8]. Octo [6] can be fine-tuned on robots with observations and actions that are distinct from those seen during pre-training. However, Octo only pre-trains on data from single robot arms and does not explore co-training on more heterogeneous data. Reed et al. [9] and Bousmalis et al. [10] propose a flexible transformer-based policy that handles varying observation and action spaces during pre-training. They demonstrate their policy can control robot arms with different action spaces, including 4-DoF, 6-DoF, 7-DoF, and 14-DoF (using a 3-finger hand). We explore the challenges in increasing both the diversity of embodiments and environments a single policy can operate in. Along with increasing the number of arm embodiments we can control, e.g., high-frequency bimanual arms, we demonstrate navigation and quadrupedal locomotion, and we evaluate across a wide range of real-world settings, while they limit their evaluation to a standardized cage. We also explicitly evaluate for transfer across embodiments by comparing policies with and without cross-embodiment co-training, while their focus is on autonomous self-improvement. Perhaps most related to our work, Yang et al. [8] study transfer across manipulation and navigation data. However, their focus is on leveraging the fact that egocentric motion in navigation looks similar to egocentric motion from wrist cameras in manipulation, and they perform manual alignment of the actions across these two embodiments. Instead, our focus is on training a policy that can control embodiments beyond arms and ground navigation robots, including those with observations and action spaces that cannot be cast to a common format. Our method, CrossFormer, is the first to co-train on four distinct action spaces (single-arm manipulation, ground navigation, bimanual manipulation, and quadrupedal locomotion) without any observation space constraints or action-space alignment while maintaining state-of-the-art performance on each robot.

Training generalist, cross-embodied policies requires multi-robot datasets, and there have been several efforts to collect such large-scale cross-embodied datasets [45, 41, 42, 5, 46]. In particular, the Open Cross-Embodiment dataset (OXE) [5] aggregates 1.5 million episodes of robot data, and we train on a subset of 900K trajectories. We note that the Octo [6] and RT-X [5] models are also trained on the OXE dataset, however they only use a subset with single robot arms. We additionally use the GNM [41] navigation and DROID [35] (large-scale Franka manipulation) datasets, along with Go1 quadruped and ALOHA [47] bimanual data collected by concurrent work [48].

## 3 Designing a Cross-Embodied Policy

The primary challenge in robotic learning with many embodiments lies in handling widely varying observation and actions spaces, as well as differences in control frequency and other aspects of the robotic system. Robotic systems can have varying numbers of camera views or proprioceptive sensors, and they may be controlled by a variety of different action representations, including joint angles, Cartesian positions, and motor torques. In order to standardize the data to a common format, some prior work on training cross-embodied policies has ignored certain observation types (such as either wrist or third person views in manipulation) [5, 7] or aligned action spaces across robots [8]. Instead, following other prior work [9, 10, 6], we cast cross-embodied imitation learning as a sequence-to-sequence problem and choose a transformer-based policy architecture that can handle varying length sequential inputs and outputs.

Due to their sequential nature, a transformer policy enables encoding all available observation types from each embodiment by serializing them into a flat sequence. Similarly, we can decode variable length actions which allows us to use the optimal action type for each embodiment. Using this flexible output, we can also predict variably sized chunks of actions. Action chunking [49, 47, 50] improves the temporal consistency of actions and reduces compounding error which is particularly important for high-frequency fine manipulation. Together, a transformer backbone and action chunking enables our policy to control robots ranging from a bimanual ALOHA system with joint position control at 20Hz, to ground and aerial navigation robots with 2D waypoint control at 5Hz.

At a high level, our transformer policy follows prior work that trains transformers on multi-modal data [9, 10, 6]. Observations and a task specification are tokenized by modality-specific tokenizers, assembled into a token sequence, and fed into a causal, decoder-only transformer backbone that is

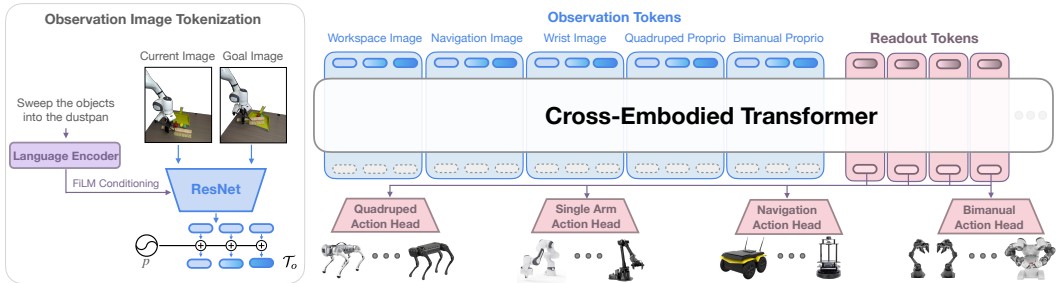

Figure 2: **Policy architecture.** Our architecture enables cross-embodied policy learning through a transformer backbone. Our policy accepts variable observation inputs by tokenizing images and proprioceptive information, predicts variable action outputs using action readout tokens, and conditions on language instructions or goal images.

shared across all embodiments. The output embeddings are then fed into separate action heads for each class of embodiment to produce actions of the corresponding dimension, as shown in Figure 2.

## 3.1 Training data

Our training data mixture covers 20 different robot embodiments, varying widely in observation space, action space, and control frequency. We start from the single-arm manipulation subset of the Open Cross-Embodiment dataset used by Octo [6]. Then, we add the DROID Franka manipulation dataset [35], 7K trajectories of ALOHA data collected across two institutions (referred to as ALOHA-multi-task), 60 hours of navigation from the GNM dataset [41], 25 minutes of walking data from a Go1 quadruped (referred to as Go1-walk), and 200 trajectories of additional Franka data collected in our own lab (referred to as Franka-tabletop). We classify the datasets that are most relevant to each of our evaluation settings (see Section 4) as the *target datasets* and up-weight them relative to the other datasets during training. Our target datasets are BridgeData [36] for WidowX evaluation, ALOHA-multi-task for ALOHA evaluation, GNM [41] for navigation evaluation, Go1-walk for quadruped eval-

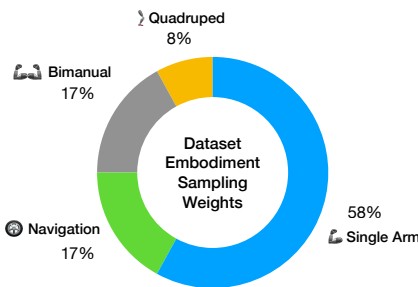

Figure 3: **Training Data Mix.** We group the 20 embodiments in our training data into categories and visualize their contribution to our data mix. The pie chart shows the average composition of each training batch based on the sampling weights.

uation, and Franka-tabletop for Franka evaluation. We collected the Go1 data by rolling out an expert policy trained with RL in simulation [51]. See Figure 3 for our training data mix and Appendix A for further details on the composition of datasets collected ourselves along with the sampling weights.

## 3.2 Tokenizing variable observation types and task specifications

The first step in training our cross-embodied policy is creating the input sequence. The trajectories in our robot training data are sequences of time-steps where each time-step contains image observations $I$, proprioceptive observations $P$, and an action. Data from each embodiment may have a different number of camera views per time-step and may or may not include proprioceptive observations. To create the input sequence, we first define an observation history length $k$ and chunk each trajectory into $k$-length segments, $[I_t, P_t..., I_{t+k}, P_{t+k}]$. We then tokenize each observation according to its modality. Images are processed with a ResNet-26 encoder [52] to produce a feature map that is flattened along the spatial dimensions and projected to the token embedding size. Proprioceptive observations are simply projected to the token embedding size. Along with the observation sequence, the policy also accepts a task specification. Importantly for cross-embodied control, our policy accepts task specifications either in the form of a language instruction $l$ or a goal image $g$. In some settings, like navigation, tasks are more naturally specified as an image goal whereas in other settings, like manipulation, tasks are more easily specified with language. Language instructions are

jointly processed with image observations using FiLM [53]. Goal images are stacked on the current image along the channel dimension before being fed into the image encoder.

Because our training data consists of data from single-arm manipulators, dual-arm manipulators, quadrupeds, and ground navigation robots, our policy supports conditioning on any subset of the following observation types: **(1) Workspace Image:** A 3rd person camera view in manipulation settings. **(2) Navigation Image:** An egocentric camera view in navigation settings. **(3) Wrist Image:** A view from a wrist-mounted camera in manipulation settings. **(4) Quadruped Proprioception:** Joint positions and velocity estimates for quadrupeds. **(5) Bimanual Proprioception:** Joint positions for bimanual manipulation settings. To maximize transfer across embodiments, we share image encoder weights for camera views of the same type. So, for example, the workspace images in single arm and bimanual manipulation settings are processed by the same ResNet image encoder. In total we use four image encoders: one for the workspace view in manipulation settings, one for the egocentric view from ground navigation robots, and two for wrist cameras in manipulation settings. After input tokenization, we have a sequence of observation tokens $[I_t^{1:L}, P_t^{1:M} ..., I_{t+k}^{1:L}, P_{t+k}^{1:M}]$, where $L$ and $M$ denote the number of tokens for image and proprioceptive observations.

### 3.3   Predicting variable length actions

After creating the input sequence, the next step is to process the input sequence with a transformer to predict an action of an appropriate dimension for each embodiment. We use a transformer with a block-wise causal attention mask such that observation tokens can only attend to observation tokens at the same or prior time-steps $t$. Following prior work [6], we insert special readout tokens $R$ after the observation tokens at each time-step in the input token sequence. These readout tokens can only attend to prior observation tokens, and thus serve as a convenient representation from which to predict actions. The final input sequence is $[I_t^{1:L}, P_t^{1:M}, R_t^{1:N} ..., I_{t+k}^{1:L}, P_{t+k}^{1:M}, R_{t+k}^{1:N}]$ where $N$ denotes the number of readout tokens. We pass the input token sequence through the transformer to obtain an embedding sequence. We then apply an action head to the embeddings corresponding to the readout tokens to produce actions. There are several possibilities for the action head. Past work has explored regression with an L1 or L2 loss, classification with a cross-entropy loss, or diffusion. We choose to predict continuous actions and employ an L1 loss because of its success in prior work on high-frequency bimanual manipulation [47]. Accordingly, our action heads simply project the readout token embeddings to the action dimension. For some embodiments, we predict a chunk of sequential actions. Action chunking has been shown to improve policy performance in prior work [47, 50] and is essential for embodiments with high control frequencies where compounding error would build up too quickly. Since our action heads project the readout tokens to the action dimension size, we match the number of readout tokens to the action chunk size for each embodiment. Our policy has 4 action heads which produce chunked actions of the following types: **(1) Single arm Cartesian positions:** A 7-dimensional action denoting the relative change in Cartesian position of the end-effector and the gripper actuation. We predict a chunk of 4 actions and execute on single robot arms at 5-15Hz [6]. **(2) Navigation waypoints:** A 2-dimensional action denoting a waypoint relative to the robot's current position. We predict a chunk of 4 actions and execute on navigation robots at 4Hz [42] . **(3) Bimanual joint positions:** A 14-dimensional action denoting the joint positions of both arms. We predict a chunk of 100 actions and execute on bimanual robot arms at 20Hz [47]. **(4) Quadruped joint positions:** A 12-dimensional action, predicted with no chunking, denoting the joint positions of the legs. We predict only 1 action and execute on a quadruped at 20Hz [51]. Action chunk sizes are taken from prior work in each robot setting.

### 3.4   Training details

In practice, we mask observations that are missing for an embodiment, so that each batch element contains all observation types and all readout token groups, and these token groups occupy fixed locations within the context window. Alternatively, for memory efficiency, observation and readout tokens could be densely packed to remove padding and fit more time-steps of context for embodiments with fewer observation types. This is a strategy used by prior work [9]. However, by not fixing observation and readout token types to a set location in the context window, the model would need to infer the embodiment purely from the observations in order to predict actions of the correct

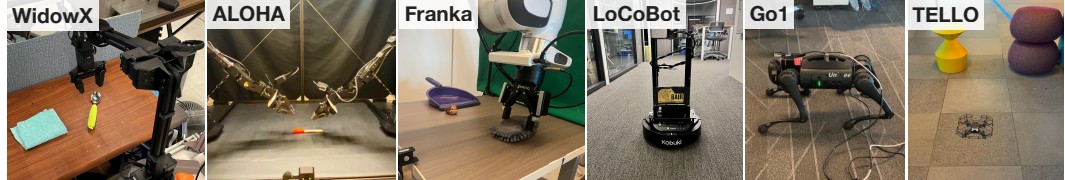

Figure 4: **Evaluation Settings:** Our tasks include single-arm manipulation settings, dexterous and bimanual task settings, navigation, and aviation. Please refer to Sec 4 for a full breakdown.

type (rather than relying on the positional embeddings of the readout tokens). The observations for some embodiments can look similar (such as navigation and manipulation with only a wrist camera), so this design may require appending a prefix to the token sequence indicating the embodiment.

Our transformer backbone has 12 layers, 8 attention heads, an MLP dimension of 2048, and a token embedding size of 512. In total, our model has 130M parameters. We initialize the ResNet-26 encoders with ImageNet pre-trained weights. We use a context widow size of 2135 tokens, which fits 5 time-steps of context with all observation and readout token groups present. We found that good performance on navigation required 5 time-steps of observation history and using this context length did not hurt performance for other embodiments. We trained for 300K gradient steps with a batch size of 512 which took 47 hours on a TPU V5e-256 pod. We use the AdamW optimizer [54], an inverse square root decay learning rate schedule [55], weight decay of 0.1, and gradient clipping of 1.0. We apply standard image augmentations. During training, we use hindsight goal relabeling and sample future observations uniformly at random to use as goals [56]. If a language instruction is available for a trajectory, we randomly mask either the language or goal so that at test time we can condition our policy using either task specification [57, 58].

## 4 Evaluation

Our experiments are designed to test whether our single cross-embodied policy can control multiple robot embodiments without sacrificing performance when compared to robot-specific imitation learning approaches. Specifically, we aim to answer the following questions: **(1)** Can our policy trained on many robots match the performance of a policy trained on data from only the target robot? **(2)** Does our policy match the performance of the best prior imitation learning method in each robot setting? **(3)** How does our policy compare to prior cross-embodied policies that align observation and action spaces?

**Evaluation setups.** We evaluate over a wide range of tasks and embodiments (see Fig. 4): **(1) WidowX Manipulation**: We use the Bridge setup from Walke et al. [36]. We use an over-the-shoulder camera view and and sample actions from the single arm head of our policy. We perform 48 trials per policy over two language-conditioned tasks and two goal-conditioned tasks. **(2) Franka Manipulation**: We use the DROID setup from Khazatsky et al. [35]. We use an over-the-shoulder camera view and sample actions from the single arm head of our policy. We perform 39 trials per policy over two language-conditioned tasks. **(3) ALOHA Bimanual Manipulation**: We use the ALOHA setup from Zhao et al. [47]. We use 3 camera views, one overhead and two wrist, and sample actions from the bimanual head of our policy. We perform 20 trials per policy over two language-conditioned tasks. **(4) LoCoBot Navigation**: We use the LoCoBot setup from Shah et al. [42] which has one camera view. We evaluate on suite of three skills: path-following, obstacle avoidance, and sharp corner-turning. We collect a topological map of goals similar to Shah et al. [42] and evaluate success based on distance to the closest node within the topological map of goal images. We sample actions from the navigation head of our policy. We evaluate in 6 locations, using one trial per location and policy as done in prior work [41, 42]. **(5) Go1 Quadruped**: We evaluate on a Unitree Go1 which uses proprioceptive observations $o_t \in \mathcal{R}^{59}$. We sample actions from the quadruped head of our policy. As our evaluation metric we report the average reward achieved over 25 minutes normalized by the the reward achieved by the RL-trained expert policy that generated the data (see Section 3.1). **(6) Tello Quadcopter**: Finally, we experiment with a Tello quadcopter using the navigation head of our policy. Since the navigation head outputs 2-D relative waypoints, we maintain a static height throughout the trajectory [42, 41]. Notably, we do not train on quadcopter

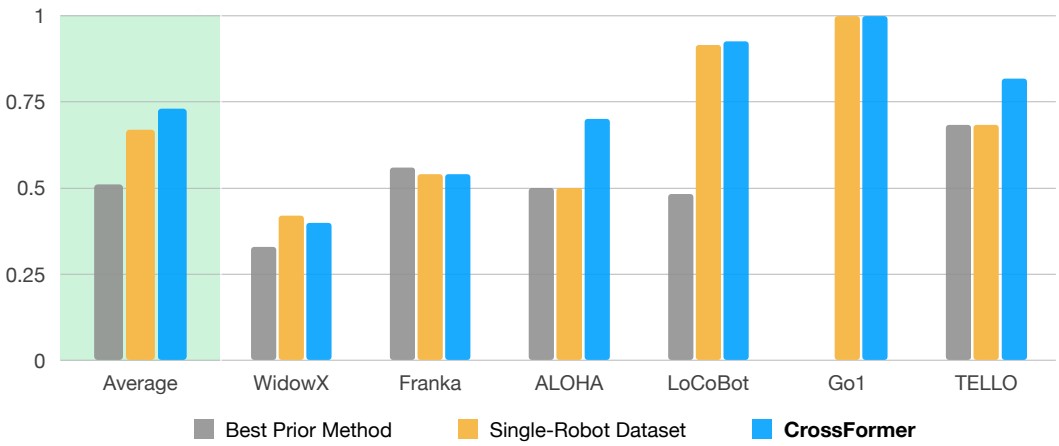

Figure 5: **Real Evaluation.** We compare CrossFormer to the same architecture trained on just the data from the target robot, as well as the best prior method on the data from the target robot.

data so this setting requires *zero-shot* generalization to a new embodiment (though not to a new set of observation inputs and action outputs). We evaluate in 3 locations, using one trial per location and policy as done in prior work [41, 42].

**Baselines and comparison methods.**    To evaluate how co-training on data from other embodiments affects performance for a given embodiment, we train our policy architecture on just the data from the target embodiment using the target dataset (see Section 3.1) for each evaluation setting. Because these datasets are smaller than our full data mix, we create smaller versions of our architecture, ranging from 5M to 95M parameters, to avoid overfitting. We refer to this method as the **single-robot dataset** baseline. We also compare our method to the **best prior method** in each setting to evaluate whether we can match the performance of an approach that was specifically tuned for each robot. For the WidowX we use Octo [6], for the Franka we use OpenVLA [59], for ALOHA we use ACT [47], for the LoCoBot and Tello quadcopter we use ViNT [42]. We do not compare to a prior state-of-the-art method for the Go1 setting due to the lack of a corresponding imitation learning method for quadrupedal locomotion. Finally, in the navigation and manipulation settings, we compare to the policy from **Yang et al. [8]**, by using the original 186M parameter policy. Yang et al. [8] leverage the fact that the egocentric view in navigation looks similar to wrist camera views in manipulation, and they align the action spaces of these two settings accordingly. This comparison evaluates whether our approach can perform similarly with no manual alignment of action spaces.

### 4.1   Can CrossFormer match the performance of training on only a single-robot dataset, as well as the best prior method in each setting?

Figure 5 shows the performance of our method compared to our architecture trained on just the single-embodiment target datasets for each setting (see Section 3.1) and the best prior method for each setting. Overall, we find that our method performs comparably to training on just the target dataset in all evaluations, demonstrating that our architecture can absorb data from widely varying embodiments with no negative transfer. On average over all embodiments, CrossFormer has a 73% success rate, while the single-robot dataset baseline has a 67% success rate. Additionally, Cross-Former performs similarly to the best prior imitation learning method in each evaluation setting, demonstrating that our policy architecture can match state-of-the-art methods that have been specifically developed and tuned for each robot. On average over all embodiments, CrossFormer has a 73% success rate, while the best prior method has a 51% success rate. Because there is no suitable prior imitation learning method for the Go1, it is not included in the reported best prior method average.

### 4.2   Is aligning the observation and action spaces really needed?

Lastly, we compare our policy to Yang et al. [8] in the LoCoBot navigation and WidowX manipulation settings. In the navigation setting, our policy significantly outperforms Yang et al. [8] in all tested scenarios including turning corners, avoiding obstacles, and making sharp turns. Qualitatively, we also found our policy to be much smoother, navigating in straighter lines with fewer stops and starts. Both policies are trained on the same amount of LoCoBot data (from the GNM dataset),

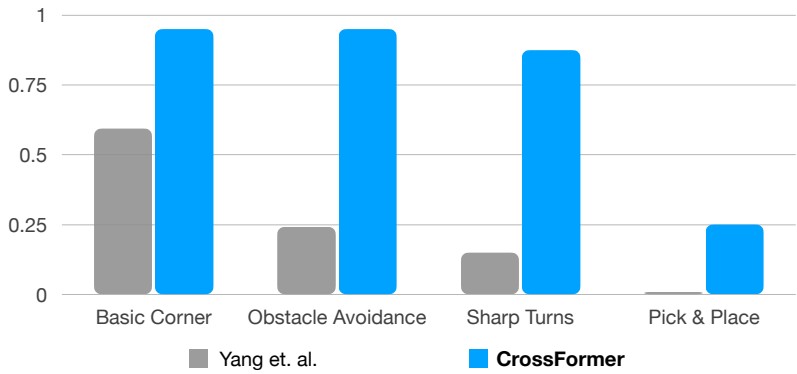

Figure 6: **Comparison to Yang et al. [8].** We compare CrossFormer to Yang et al. [8], which aligns actions for navigation and manipulation and only uses a single camera view at a time. CrossFormer outperforms Yang et al. [8] by 3x overall, on both tabletop manipulation from a third person camera view as well as common navigation tasks.

so this result indicates that our policy architecture can better fit its cross-embodiment training data. Surprisingly, in the WidowX manipulation setting, we found that Yang et al. [8] obtained a zero success rate, despite training on the same relevant WidowX data as CrossFormer (i.e the Bridge dataset) and additional WidowX data collected in their own lab. We hypothesize that this is because we evaluate from the 3rd-person camera view. While Yang et al. [8] train with both 3rd-person and wrist camera views, only one view is input to the policy at a time, potentially resulting in underfitting to 3rd-person control. CrossFormer only uses the 3rd-person view for the WidowX setting due to a lack of publicly available WidowX wrist-camera data. However, we note that our policy can accept a varied number of camera views (using up to three simultaneously for bimanual manipulation) and does not require forcing two camera views into the same input slot. In summary, our policy outperforms Yang et al. [8] in both settings, demonstrating that manual alignment of observation and action spaces is not necessary for good performance, and that our flexible architecture allows our policy to better fit heterogeneous cross-embodied data.

## 5 Discussion and Conclusion

We introduced CrossFormer, a scalable and flexible transformer policy trained on the largest and most diverse dataset to date, 900K trajectories across 20 different robot embodiments. We demonstrated a principled way to learn a single policy that can control vastly different embodiments including single and dual arm manipulation systems, wheeled robots, quadcopters, and quadrupeds. Our results show that CrossFormer matches the performance of specialist policies tailored for individual embodiments, while also significantly outperforming the state of the art in cross-embodiment learning. However, our work does have limitations. Our results do not yet show significant positive transfer across embodiments. We anticipate that as we train on larger robot datasets with more embodiments, we will see greater positive transfer. Another limitation is that our data mix uses hand-picked sampling weights to avoid over-training on datasets with many repetitive episodes and under-training on the data most relevant to our evaluation settings. In principle, as we scale model size, the policy should have the capacity to fit all the data equally well without any data weighting. Finally, given that we need large models to fit large multi-robot datasets, the model's inference speed can become a limiting factor. In this work we successfully applied our policy to a high-frequency, fine-grained bimanual manipulation task, but as we scale the model's size we may not be able to control these higher frequency embodiments. Future hardware improvements will help to alleviate the issue, but further research is needed on techniques for using large models to control high-frequency robots. Future work could also include exploring techniques to enable greater positive transfer across embodiments while maintaining the flexibility of our architecture, techniques for data curation, and incorporating even more diverse data sources like sub-optimal robot data or action-free human video. We hope that this work will open the door for more general-purpose and flexible robot policies that can efficiently learn and transfer knowledge from the experience collected across diverse robot embodiments.

**Acknowledgments**

We thank Laura Smith, Kyle Stachowicz, Karl Pertsch, Ajay Sridhar, Jonathan Yang, and Catherine Glossop for help with setting up the robots and baselines. This research was supported by the TPU Research Cloud, NSF IIS-2150826, ARL DCIST CRA W911NF-17-2-0181, and ONR N00014-20-1-2383.

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

More information and videos can be found on our website: crossformer-model.github.io.

## A  Training Data

We list the sampling weights for each dataset in our data mixture in Table 1. We up-weight our target datasets: Bridge, ALOHA-multi-task, GNM, Go1-walk, and Franka-tabletop.

| CrossFormer Dataset Mixture | |
| --- | --- |
| Fractal [32] | 17% |
| Kuka [28] | 2.2% |
| BC-Z [60] | 2.2% |
| Stanford Hydra Dataset [61] | 0.015% |
| Language Table [62] | 1.5% |
| Taco Play [63, 64] | 1.2% |
| Furniture Bench Dataset [65] | 0.83% |
| UTAustin Mutex [66] | 0.76% |
| Austin Sailor Dataset [67] | 0.74% |
| Roboturk [68] | 0.79% |
| Toto [69] | 0.68% |
| Austin Sirius Dataset [70] | 0.59% |
| Berkeley Autolab UR5 [71] | 0.41% |
| IAMLab CMU Pickup Insert [72] | 0.31% |
| Viola [73] | 0.32% |
| Berkeley Fanuc Manipulation [74] | 0.26% |
| NYU Franka Play Dataset [75] | 0.28% |
| Jaco Play [76] | 1.6% |
| Berkeley Cable Routing [77] | 0.089% |
| Austin Buds Dataset [78] | 0.072% |
| CMU Stretch [79] | 0.053% |
| DLR EDAN Shared Control [80] | 0.019% |
| DROID [35] | 0.022% |
| Bridge [37, 36] | 17% |
| GNM [41] | 17% |
| ALOHA-multi-task | 17% |
| Go1-walk | 8.5% |
| Franka-tabletop | 8.5% |

Table 1: The CrossFormer training data mixture uses datasets from the Open X-Embodiment dataset [5] and additional data collected for this project.

## B  Training Hyperparameters

In Table 2 we list the hyperparameters for the optimizer and policy architecture. In total, along with the ResNet-26 image encoders and action heads, our model has 130M parameters. We initialize the ResNet-26 encoders with ImageNet pre-trained weights. Training took 80 hours on a TPU V5e-256 pod. We apply color jitter and random resizing/cropping image augmentations. During training, we use hindsight goal relabeling and sample future observations uniformly at random to use as goals [56]. If a language instruction is available for a trajectory, we randomly mask either the language or goal so that at test time we can condition our policy using either task specification [57]. For our method and baselines we trained, we selected checkpoints based on the validation mean squared error since we found this very roughly correlates with robot performance.

## C  Evaluation Setups

Below we provide further details on our evaluation settings (see Fig. 4 for images). For goal-condtioned tasks, we arrange the scene and take the goal image before rolling out the policy. For the language-conditioned tasks, we use language instructions that are semantically in the distribution of the training data (though may not appear exactly in the training data).

| Hyperparameter | Value |
|---|---|
| Optimizer | AdamW [54] |
| Learning Rate | 3e-4 |
| Warmup Steps | 2000 |
| LR Scheduler | reciprocal square-root |
| Weight Decay | 0.1 |
| Gradient Clip Threshold | 1 |
| Batch Size | 512 |
| Layers | 12 |
| Attention heads | 8 |
| Token embedding size | 512 |
| MLP dimension | 2048 |
| Context length | 2135 |
| Total training steps | 300K |

Table 2: Training hyperparameters for CrossFormer.

For the manipulation settings, we ran the models on a machine with an NVIDIA 4090 (or similar GPU) that was connected to the robot. For the ground navigation and drone experiments, we ran the models on a separate server and sent actions over a wireless network.

**WidowX Manipulation**    We use the Bridge setup from Walke et al. [36]. We use an over-the-shoulder camera view and and sample actions from the single arm head of our policy. We evaluated two language-conditioned tasks, putting a spoon on a cloth and putting a carrot on a plate, and two goal-conditioned tasks, putting a mushroom in a pot and putting a cloth on a saucer. We used 12 trials per policy and task. Positions of the objects were varied between trials.

**Franka Manipulation**    We use the DROID setup from Khazatsky et al. [35]. We use an over-the-shoulder camera view and sample actions from the single arm head of our policy. We evaluated for 27 trials per policy on the language-conditioned task of using a sponge to sweep pinecones into a dustpan. We also evaluated for 12 trials per policy on the task of flipping a pot upright. The positions of the objects were varied between trials.

**ALOHA Bimanual Manipulation**    We use the ALOHA setup from Zhao et al. [47]. We use three camera views, one overhead and two wrist, and sample actions from the bimanual head of our policy. We evaluated two language-conditioned tasks, taking the cap off of a pen and picking up a knife and using it to cut a piece of sushi. We used 10 trials per policy and task. The position of the pen, knife, and sushi was varied between trials.

**LoCoBot Navigation**    We use the LoCoBot setup from Shah et al. [42] which has one camera view. We evaluate on suite of three skills: path-following, obstacle avoidance, and sharp corner-turning. We sample actions from the navigation head of our policy. We combine our policy with the graph-based planner and distance function from Shah et al. [42]. We first obtain a topological map $\mathcal{M}$ of the environment by teleoperating the robot. Then, at every time-step we find the closest node in $\mathcal{M}$, search the graph for the shortest path from this node to the goal, and command the policy with the most immediate subgoal in the path. We evaluate the success of a trajectory based on the number of subgoals between the closest node at the end of the trajectory and the goal. We evaluate in 6 locations, using one trial per location and policy as done in prior work [41, 42].

**Go1 Quadruped**    We evaluate on a Unitree Go1 which uses proprioceptive observations $o_t \in \mathcal{R}^{59}$. We sample actions from the quadruped head of our policy, and the task is to walk forward. Importantly, unlike prior work [8], we directly control the quadruped's joints rather than doing higher level control with navigation waypoints. As our evaluation metric we report reward achieved

| | Task | Single-Robot Dataset | Best Prior Method | CrossFormer |
|---|---|---|---|---|
| WidowX | Put the spoon on the cloth | 0.25 (12) | 0.25 (12) | 0.25 (12) |
| | Put the mushroom in the pot | 0.00 (12) | 0.17 (12) | 0.25 (12) |
| | Put the cloth on the saucer | 0.75 (12) | 0.67 (12) | 0.75 (12) |
| | Put the carrot on the plate | 0.67 (12) | 0.25 (12) | 0.33 (12) |
| Franka | Sweep the pinecones into the dustpan | 0.41 (27) | 0.52 (27) | 0.41 (27) |
| | Flip the pot upright | 0.83 (12) | 0.67 (12) | 0.83 (12) |
| ALOHA | Uncap the pen | 0.60 (10) | 0.70 (10) | 0.80 (10) |
| | Use the knife to cut the sushi | 0.40 (10) | 0.30 (10) | 0.60 (10) |
| LoCoBot | Obstacle avoidance | 0.95 (2) | 0.30 (2) | 0.95 (2) |
| | Cornering | 0.95 (2) | 0.85 (2) | 0.95 (2) |
| | Sharp cornering | 0.85 (2) | 0.30 (2) | 0.88 (2) |
| Tello Quadcopter | Cornering | 0.68 (3) | 0.68 (3) | 0.82 (3) |
| Go1 | Walking | 1 | N/A | 1 |
| Average | | 0.68 (116) | 0.51 (116) | 0.73 (116) |

Table 3: **Real Evaluation (detailed).** We compare CrossFormer to the same architecture trained on just the data from the target robot, as well as the best prior method on the data from the target robot. The number of trials is in parentheses. For ground and aerial navigation (LoCoBot and Tello Quadcopter) the success metric is the proportion of subgoals reached on the path towards the goal. For the Go1, we report reward achieved over 25 minutes normalized by the reward achieved by the RL-trained expert policy that generated the data. Because there is no suitable best prior imitation learning method for the Go1, we do not include it in the best prior method average.

over 25 minutes normalized by the reward achieved by the RL-trained expert policy that generated the data (see Section 3.1).

**Tello Quadcopter** Finally, we perform evaluation on a Tello quadcopter using the navigation head of our policy. Since the navigation head outputs 2-D relative waypoints, we maintain a static height throughout the trajectory [42, 41]. Notably, we do not train on quadcopter data so this setting requires *zero-shot* generalization to a new embodiment. We evaluated on the skill of turning a corner. We evaluated in 3 locations, using one trial per policy and location as done in prior work [41, 42].

## D   Evaluation Results

In Table 3 we provide a more detailed breakdown of our evaluation results by task.

