# OpenReview forum: "Scaling Cross-Embodied Learning: One Policy for Manipulation, Navigation, Locomotion and Aviation"
_robot-learning.org/CoRL/2024/Conference — CoRL 2024_

### Official Review · Reviewer_ZGgs · 2024-07-21
**Learning a single policy for manipulation, locomotion and aviation**

**Originality:** 4
**Technical Quality:** 4
**Clarity Of Presentation:** 3
**Potential Impact:** 4
**Recommendation:** 4
**Confidence:** 5

**Review:**

This paper introduces CrossFormer, a scalable and flexible transformer-based policy designed to control 30 different types of robots, each equipped with varying sensors, actuators, and control frequencies. The robots encompass single and dual-arm manipulation systems, wheeled robots, quadcopters, and quadrupeds. CrossFormer is a transformer-based policy that supports variable observations and actions by converting inputs and outputs into sequences. This approach allows for the training of robot data with any number of camera views or proprioceptive sensors by tokenizing and arranging the observations into a sequence. Although this cross-embodied imitation learning method can handle a wide range of embodiments, it does not achieve the highest performance compared to specialized policies. For instance, reinforcement learning (RL)-trained policies for locomotion outperform the trained policy of this work in terms of optimality, robustness, agility, etc.

Major comments:
- The proposed method excels in cross-embodiment learning, outperforming the state-of-the-art in this area, which is commendable. However, it still falls short compared to specialized policies in certain cases. This raises an important question: should we aim for a cross-embodiment and cross-task policy using a single policy, or should we develop a unique policy for each task and embodiment? For example, the locomotion performance of this work should be compared with the following study that focuses on cross-embodiments with only the locomotion task, which seems to exhibit much better locomotion behavior than the A1 robot proposed in this paper:

Shafiee, Milad, Guillaume Bellegarda, and Auke Ijspeert. "Manyquadrupeds: Learning a single locomotion policy for diverse quadruped robots." arXiv preprint arXiv:2310.10486 (2023).

- Additionally, the sim-to-real transfer of quadruped robots requires attention. The current simulation for locomotion needs improvement as it exhibits unnatural behavior, with legs making unnecessary contact with the ground during normal locomotion tasks, which have already been solved.

- Another consideration is the energy efficiency of tasks. If the collected data is not efficient and optimal, how can the method address and improve it?

- The potential of cross-embodiment policies is significant for researchers aiming to control and train their own embodiments. Do you plan to open-source your training framework? This would substantially increase the impact of your current work.

- A frequently asked question is, if we already have data from a working robot, meaning the task is solved, why should we use this data again to train a generalist policy?

- "Does our policy match the performance of the best prior imitation learning method in each robot setting?" This is an interesting question addressed in the paper. Another crucial question is whether the proposed policy matches the performance of the best prior reinforcement learning method in each robot setting.

- One limitation that should be addressed is that the current framework can only learn tasks that have already been solved. How will your future work address unseen tasks that have not yet been solved? This discussion should be included in the limitations section.

Minor comments:
- Refer to Figure 3 for an overview of our architecture --> should be Figure 2.
- Figure 3: Training Data,  has not been cited in the text of the paper.

**Quality Of The Limitations Section:**

2

**Questions For Rebuttal:**

Please check the above comments for the questions.

**Robotics Focus:**

4

**Summary Of Paper:**

This paper introduces CrossFormer, a scalable and flexible transformer-based policy designed to control 30 different types of robots, each equipped with varying sensors, actuators, and control frequencies. The robots encompass single and dual-arm manipulation systems, wheeled robots, quadcopters, and quadrupeds. CrossFormer is a transformer-based policy that supports variable observations and actions by converting inputs and outputs into sequences. This approach allows for the training of robot data with any number of camera views or proprioceptive sensors by tokenizing and arranging the observations into a sequence. Although this cross-embodied imitation learning method can handle a wide range of embodiments, it does not achieve the highest performance compared to specialized policies. For instance, reinforcement learning (RL)-trained policies for locomotion outperform the trained policy of this work in terms of optimality, robustness, agility, etc.

**Summary Of Recommendation:**

The paper addresses one of the most significant challenges in robotics learning: developing a cross-embodiment and cross-task learning framework. It presents promising results both in hardware and simulation. Therefore, I recommend accepting the paper.

---

### Official Review · Reviewer_Cs6U · 2024-07-21
**A novel flexible-observation policy representation that enables training on a larger cross-embodiment dataset.**

**Originality:** 3
**Technical Quality:** 3
**Clarity Of Presentation:** 3
**Potential Impact:** 4
**Recommendation:** 3
**Confidence:** 4

**Review:**

The paper is inspired by Yang et al.[1] work on cross-embodiment learning for Manipulation and Navigation. The use of a substantial dataset that spans multiple embodiments and tasks is a significant strength. By avoiding the need for manual alignment of observation and action spaces, the method simplifies the training process and reduces the potential for human error.

There is a novelty in the presented work, but the results are lacking and need much more analysis and details.

Criticism:
The evaluation tasks lack sufficient description in manipulation. The choice of using multiple trials on just “a language conditioned” and “a goal-conditioned” (line 245) task is questionable. The website consists of a single task per embodiment. If this is a typo and you do have multiple tasks, each with a different language/goal condition, then please correct it. Further, these tasks need more description explaining what they are and each task needs to have multiple trials.

Analysis in different settings for navigation. Yang et al provided a detailed analysis of their method in different settings/locations. This detail is missing in the author's results, making it weaker.

Although the paper does analyze the performance of their architecture trained on single policies, an ablation of different combinations of the data mixture would be extremely helpful (as was done in Yang et al). For example, is the quadruped data inconsequential in the presence of navigation data?

The authors hypothesize their mode works better on navigation than the baseline because their method involves training with a flexible observation space. This hypothesis can be tested with another training run where the policy is just shown one of the camera views during training. Please add this ablation result as well.

An ablation of the model capacity would be helpful as well.


References:
[1]: Yang, Jonathan, et al. "Pushing the limits of cross-embodiment learning for manipulation and navigation." arXiv preprint arXiv:2402.19432 (2024).

**Quality Of The Limitations Section:**

3

**Questions For Rebuttal:**

For each robot, can you describe how the model was deployed on the real robot, for example, what hardware was used to fit a 110M parameter model onto each of these embodiments? Was it controlled via a laptop/cloud or was it deployed on the actual robot?

**Robotics Focus:**

4

**Summary Of Paper:**

The paper introduces CrossFormer, a transformer-based policy designed to manage diverse robotic tasks across various embodiments, such as single and dual arm manipulation, wheeled robots, quadcopters, and quadrupeds. It leverages a large and varied dataset (900K trajectories across 30 different robot embodiments) to achieve generalized control, without the need for manual alignment of observation or action spaces. The method aims to match the performance of specialized policies while outperforming previous state-of-the-art approaches in cross-embodiment learning.

**Summary Of Recommendation:**

The paper proposes a novel model architecture for a larger dataset. The paper mentions that their results are over one task in each embodiment. This is not a strong enough result and needs more analysis.

---

### Official Review · Reviewer_xtGM · 2024-07-21
**A good, well-written paper presenting a solid step forward towards an effective cross-embodiment learning architecture for robotics based on the Transformer architecture.**

**Originality:** 3
**Technical Quality:** 4
**Clarity Of Presentation:** 4
**Potential Impact:** 3
**Recommendation:** 4
**Confidence:** 4

**Review:**

The paper presents an architecture called CrossFormer that is capable of integrating diverse robot body types and observation and action spaces. The paper is very well written. Overall, it address the following question: how to reduce the need for hand-designed mappings of observations to and actions from a cross-embodiment policy model such that what remains is justifiably minimal. Given the choice of a Transformer architecture for implementing the policy model, that question becomes how to map into observation tokens and from action tokens. The proposed methods (building on existing work) were shown to cover navigation, dexterous manipulation and quadruped locomotion with robot body types ranging from arms to legged robots to drones. While the specific technical design still leaves much room for further improvement (e.g. positive transfer across body types is yet to be demonstrated; see also the Limitations section and Questions for Rebuttal below), the technical quality is high and the contribution is timely. The empirical evaluation on real robots is compelling. It is likely going to stimulate follow-up work in this direction that is at least partly based on the work presented here.

**Quality Of The Limitations Section:**

1

**Questions For Rebuttal:**

1. As of now, the information provided is inadequate for producing the work. For example, the specific values of the parameters in the tokenization scheme (L, M, N) and action token projection details etc are missing. Please consider including these in the main text, appendix, or open-sourced code (if code will be released).

2. Please bring the Limitations sectoin into the main text. The discussion there is highly valuable. Also, in discussing limitations of the current approach, there is no need to reiterate the merit of CrossFormer (e.g. the sentence starting on Line 560). Yet another limitation that is worth discussing is the situation with zero padding: How pervasive is zero padding in observation tokenization? How could that negatively affect model efficiency or performance? Should we consider possible architectures that entirely do away with zero padding?

3. Typo: Line 132 "Figure 3" should be "Figure 2".

4. The bald claim on Lines 285-286: "On average over all embodiments, CrossFormer has a 70% success rate while the best prior method has a 51.8% success rate" should be unpacked. For example, given Figure 5, it seems that "the best prior method" scores 0 on Go1, which presumably contributed a lot to the 70% vs. 51.8% success rate gap. The situation here should be explained.

5. More discussion on the conceptual underpinnings of the design choices could help, naturally without increasing the length of the paper.

**Robotics Focus:**

4

**Summary Of Paper:**

The paper (1) builds on prior work on cross-embodiment in robot learning, such as the Open Cross Embodiment dataset, the use of readout token for action representation in Transformer-based sequence modeling, (2) proposes a tokenization scheme that covers both image and proprioceptive observations, (3) proposes an action token decoding scheme that covers navigation, manipulation, and quadruped locomotion, (4) includes data from additional robot embodiment types, and (5) demonstrates successfully that the overall scheme supports imitation learning of a single policy for an unprecedently diverse set of robot embodiments with competitive performance.

**Summary Of Recommendation:**

A very well written paper with solid technical contribution in a direction (cross-embodiment robot learning) that is timely. The real-robot demonstrations are compelling. The results are believable. It is likely going to stimulate follow-up work in this direction that is at least partly based on the work presented here.

---

### Official Review · Reviewer_MK5T · 2024-07-30
**Review on Crossformer**

**Originality:** 3
**Technical Quality:** 5
**Clarity Of Presentation:** 4
**Potential Impact:** 3
**Recommendation:** 4
**Confidence:** 4

**Review:**

Strength:

With the introduction of large robotic datasets such as OXE, it became vital for the robot learning field to develop Large Models that can leverage these large datasets. The paper introduces a model that tackles this, and this is why I believe this paper is important. Furthermore, according to the results of the evaluations the method’s performance increases (or does not drop) when additional data from other embodiments are used during training. The model also significantly outperforms the baseline (Yang et al. [8]) on navigation and manipulation tasks.

Weakness:
 - The limitations section should not be in the supplementary.
 - There are some design decisions in the paper that are not justified properly like the chunk sizes for different actions.
 - Figure 5 includes the comparison to baseline methods and includes both single robot dataset and mixed dataset results and that result is named crossformer. In Figure 6, there is only crossformer is used in the legends. While it is explained in section 4.2, it creates confusion. The fact that the text does not state in 4.2 which dataset is used to train the models for WidowX comparison adds to the confusion.
 - The number of trials in the experiments seems a little too low.
 - How the language conditioning part of the system works or how it is utilized during evaluations is not properly explained.
 - The manuscript explains the training details comprehensively. Yet, it only states that the training took 900K steps. I would like to see how the model performance changes as the training progresses. How do you deal with overfit/underfit?
 - The evaluations of bi-manual manipulation use the single-arm action head which is confusing considering there is an action head in the model for bi-manual control.
 - I don’t think using 2D navigation for Tello and stating it as evidence of zero-shot generalization to a new embodiment is appropriate. With the same logic, since it uses cartesian position change of end effector, the single-arm head can be used for generalization to almost any single-arm manipulator which can all be considered as new embodiments.
 - From what the paper states, the performance of the model for a robot type increases if there is useful information in the data of the other robot types like wrist camera information of single-arm robots being useful for navigation and vice versa. However, based on the given results, this is a bit speculative. I suggest you include an additional analysis where the data of the others are added to the data mix gradually so that we can see adding the data of which robot to the mix increases the performance of which robot.

**Quality Of The Limitations Section:**

3

**Questions For Rebuttal:**

- Can you explain the language conditioning?
 - Can you provide details on how the model performance changes during different checkpoints in training?
 - About using the single arm action head for bi-manual evaluation, the most significant increase between using single robot data and using all data is seen in the ALOHA evaluation, Figure 3 shows that only 17 percent of the data is from bimanual robots (appendix shows that all of it is from ALOHA), since during the evaluation single arm head is used, and the remaining single-arm data is the largest portion of the data, could that be the reason in the significant increase in the evaluation success? Would we be seeing a similarly significant increase in performance if the bi-manual action head was used? Please explain.
 - One question I have is about the motivation. From what I understand, the motivation is to create a single model for all robot embodiments so that they can help each other’s performance. However, some are completely unrelated to each other and as the paper states, don’t help each other at all like quadruped and navigation. So, why use them together in the same model? As the limitations section states, as the model grows larger, the latency will increase and as we add more data from more embodiments to the data mix the model will have to grow to learn to them, so using them together can also be detrimental to model performance. Please explain.
 - The limitations section states “CrossFormer does not require any additional engineering to add data from new embodiments with different observation or action types, making scaling the training data straightforward.” How can you add the data of a humanoid robot?
 - In section 4.2, you state “We hypothesize that this is because we evaluate from the 3rd person camera view. While Yang et al. [8] train with both 3rd person and wrist camera views, only one view is input to the policy at a time, potentially resulting in underfitting to 3rd-person control.” How do the results change when the models are trained using only one view? Does the proposed model still outperform the baseline? You also state in section 4.2 “In comparison, our policy is not bound to a fixed observation space and can jointly train with 3rd-person views, a wrist-camera view, both, or neither”, yet in section limitations, it is stated that, “Another limitation is that our data mix uses hand-picked sampling weights to avoid over-training on datasets with many repetitive episodes and under training on the data most relevant to our evaluation settings.” so without the handpicked weights your model would be over trained too? Please explain
 - You state that you can alleviate this problem of hand-picked weights using model scaling. Yet, the models you used for comparison can potentially outperform your model with model scaling. What are the sizes of the models you used for baseline comparison methods?

**Robotics Focus:**

4

**Summary Of Paper:**

The paper focuses on leveraging information collected by/for other robots to increase the performance of other robots and introduces a method called Crossformer. The method is evaluated using several different settings using several different robots with various morphologies against other state-of-the-art methods. The evaluations also include real-world testing.

**Summary Of Recommendation:**

I find the method and the motivation interesting. The evaluation is a bit weak, the method description is slightly lacking, and some parts of the model are not used in any of the evaluations (bimanual output).

---

### Author Rebuttal · Authors · 2024-08-10

We thank all the reviewers and AC for their thoughtful comments, and we are glad to see that our work was received positively! We have responded to each reviewer individually below. To summarize, we added additional tasks to strengthen our evaluation and revised the paper to further clarify and justify details of the method. Please see the attached PDF for the revisions (highlighted in blue).

**Additional Evaluation Experiments**

As requested by the reviewers, we have added additional tasks to the evaluation of our method and the baselines. We added two new WidowX tasks with 12 trials per policy, one new Franka task with 12 trials per policy, and one new ALOHA task with 10 trials per policy. Now, our primary comparison consists of 13 tasks across 6 embodiments with a total of 116 trials per method (348 trials total). The conclusion of our evaluation remains the same: CrossFormer performs comparably or better than policies trained on only data from the target robot, as well as the best prior imitation learning method for each robot setting.

---

### Decision · Program_Chairs · 2024-09-04

**Decision:**

Accept

**Comment:**

The paper presents a sound and interesting method for the challenging problem of integrating diverse robot (arms, legged robots, drones) sensorimotor spaces. The method outperforms the baselines, is evaluated with compelling real-robot experiments, and provides scalability. The paper can be improved by addressing the reviewers’ comments about the missing details in the methods and the experimental evaluations.

The problem is challenging and timely, the method is sound, the approach is verified in various systems, and the results are impressive. I recommend accepting the paper as oral presentation.